

# Identification of key genes, pathways and potential therapeutic agents for liver fibrosis using an integrated bioinformatics analysis

Zhu Zhan[1,2], Yuhe Chen[1,2], Yuanqin Duan[1,2], Lin Li[3], Kenley Mew[4], Peng Hu[1,2], Hong Ren[1,2] and Mingli Peng[1,2]

[1] Key Laboratory of Molecular Biology for Infectious Diseases (Ministry of Education), Chongqing Medical University, Chongqing, China
[2] Department of Infectious Diseases, The Second Affiliated Hospital of Chongqing Medical University, Chongqing, China
[3] Department of Hepatic Disease, Chongqing Traditional Chinese Medicine Hospital, Chongqing, China
[4] Department of Foreign Language, Chongqing Medical University, Chongqing, China

Corresponding author
Mingli Peng,
Peng_mingli@hospital.cqmu.edu.cn

## ABSTRACT

**Background**. Liver fibrosis is often a consequence of chronic liver injury, and has the potential to progress to cirrhosis and liver cancer. Despite being an important human disease, there are currently no approved anti-fibrotic drugs. In this study, we aim to identify the key genes and pathways governing the pathophysiological processes of liver fibrosis, and to screen therapeutic anti-fibrotic agents.

**Methods**. Expression profiles were downloaded from the Gene Expression Omnibus (GEO), and differentially expressed genes (DEGs) were identified by R packages (Affy and limma). Gene functional enrichments of each dataset were performed on the DAVID database. Protein–protein interaction (PPI) network was constructed by STRING database and visualized in Cytoscape software. The hub genes were explored by the CytoHubba plugin app and validated in another GEO dataset and in a liver fibrosis cell model by quantitative real-time PCR assay. The Connectivity Map L1000 platform was used to identify potential anti-fibrotic agents.

**Results**. We integrated three fibrosis datasets of different disease etiologies, incorporating a total of 70 severe (F3–F4) and 116 mild (F0–F1) fibrotic tissue samples. Gene functional enrichment analyses revealed that cell cycle was a pathway uniquely enriched in a dataset from those patients infected by hepatitis B virus (HBV), while the immune-inflammatory response was enriched in both the HBV and hepatitis C virus (HCV) datasets, but not in the nonalcoholic fatty liver disease (NAFLD) dataset. There was overlap between these three datasets; 185 total shared DEGs that were enriched for pathways associated with extracellular matrix constitution, platelet-derived growth-factor binding, protein digestion and absorption, focal adhesion, and PI3K-Akt signaling. In the PPI network, 25 hub genes were extracted and deemed to be essential genes for fibrogenesis, and the expression trends were consistent with GSE14323 (an additional dataset) and liver fibrosis cell model, confirming the relevance of our findings. Among the 10 best matching anti-fibrotic agents, Zosuquidar and its corresponding gene target ABCB1 might be a novel anti-fibrotic agent or therapeutic target, but further work will be needed to verify its utility.

![PeerJ]

**Conclusions**. Through this bioinformatics analysis, we identified that cell cycle is a pathway uniquely enriched in HBV related dataset and immune-inflammatory response is clearly enriched in the virus-related datasets. Zosuquidar and ABCB1 might be a novel anti-fibrotic agent or target.

## INTRODUCTION

Hepatic fibrosis is characterized by the pathological accumulation of extracellular matrix (ECM) following chronic liver injury arising from various sources including toxic damage, viral infections, autoimmune conditions, and metabolic or genetic diseases. Patients with advanced liver fibrosis generally have a poor prognosis as they often develop decompensated cirrhosis and hepatocellular carcinoma (*Tsochatzis, Bosch & Burroughs, 2014*).

For the management of patients with hepatic fibrosis in clinical practice, several well-validated clinical practice guidelines and recommendations have been established, including antiviral therapy for patients with chronic hepatitis B or C (*Liver, 2017a*; *Liver, 2017b*), a cessation of alcohol consumption in patients with alcoholic liver disease (*Liver, 2012*), and lifestyle modifications in patients with nonalcoholic fatty liver disease (*Djordjevic et al., 2018*). However, eliminating the cause of fibrosis is not generally sufficient to halt progression from liver fibrosis to cirrhosis (*Feng et al., 2018*). Unfortunately, there are no approved anti-fibrotic drugs currently available (*Bottcher & Pinzani, 2017*). A better understanding of the molecular mechanisms controlling the fibrotic response is thus needed to facilitate the development of new drugs, and to thereby improve patient outcomes.

High-throughput sequencing technology offers an ideal means of profiling large gene expression datasets in order to gain a comprehensive understanding of the mechanisms underlying fibrosis. For example, *Chan et al. (2016)* found that in cirrhotic liver tissues there is a unique gene expression pattern related to inflammation, the immune response, and cell growth, and with a potential relationship with cancer as well. Using a bioinformatics analysis, many hub genes which are essential to fibrogenesis have been identified. For example, ITGBL1 was identified in an HBV-related fibrosis dataset (*Wang et al., 2017*). LUM, THBS2, FBN1, and EFEMP1 were all identified in a NAFLD-related fibrosis dataset (*Lou et al., 2017*). TAF1, HNF4A and CALM2 were identified in an HCV-related fibrosis dataset (*Ji et al., 2018*). COL6A1, COL6A2, COL6A3, PIK3R, COL1A1, and CCND2, were identified in NAFLD and HCV-related datasets (*Chen et al., 2017*). However, it remains unclear as to whether these pathways and hub genes are unique to individual disease etiologies or are shared between them.

In order to clarify this uncertainty, we integrated three datasets, each pertaining to fibrosis of a different etiology. Using bioinformatics analyses, we thereby sought to identify key genes and pathways of interest, and to screen for therapeutic agents and novel targets with the potential to treat liver fibrosis.

**Table 1  Accession information for datasets downloaded from the GEO database.** GSE6764, GSE49541 and GSE84044 was used for identifying DEGs; GSE14323 was used for validation.

| Accession | GPL | Etiology | Sample size case/control | Sample fibrosis stage | Country | Year |
|---|---|---|---|---|---|---|
| GSE6764 | GPL570 | HCV | 10/13 | F4/FO | USA | 2007 |
| GSE49541 | GPL570 | NAFLD | 32/40 | F3–F4/F0–F1 | USA | 2013 |
| GSE84044 | GPL570 | HBV | 28/63 | F3–F4/F0–F1 | China | 2016 |
| GSE14323 | GPL571 | HCV | 41/9 | F4/F0 | USA | 2009 |

## MATERIALS & METHODS

### Microarray data

Four gene expression datasets were downloaded from the Gene Expression Omnibus (GEO) database; three were analyzed to identify DEGs, while one was used for validation. Table 1 summarizes the pertinent information for the selected GEO datasets used in this study. GSE6764 (*Wurmbach et al., 2007*), GSE49541(*Moylan et al., 2014*), and GSE84044 (*Wang et al., 2017*) represent datasets from patients with liver fibrosis arising from hepatitis C virus (HCV), nonalcoholic fatty liver disease (NAFLD), and hepatitis B virus (HBV), respectively. All three of these gene expression profiles were based on the GPL570 platform. GSE49541 and GSE84044 were derived from two liver fibrosis studies in which tissues with severe fibrosis (F3–F4) and mild fibrosis (F0–F1) were selected. GSE6764 and GSE14323 (*Mas et al., 2009*) (used for validation) were derived from two liver cancer studies, in which cirrhotic (F4) and normal tissues (F0) were selected.

### Identification of differentially expressed genes

Background expression value correction and data normalization were conducted for the raw data in each dataset using an R package (Affy, version 1.52.0). Probes in each data file were then annotated based on the appropriate platform annotation files. Probes without matching gene symbols were removed. In instances where different probes mapped to the same gene, the mean value of all probes mapping to that gene was taken as the final expression value for that gene. Then, the Linear Models for Microarray Analysis R package (limma; version 3.30.11; *Ritchie et al. (2015)*) was applied for differential expression analysis. Those genes with an adjusted $P$-value < 0.05 and absolute value of fold-change (FC) >1.5 were deemed to be the DEGs. DEGs overlapping between datasets were obtained using an online Venn analysis tool (http://bioinformatics.psb.ugent.be/webtools/Venn/).

### Gene Ontology and pathway enrichment analyses

Gene Ontology (GO) is a commonly used bioinformatics tool that provides comprehensive information on gene function of individual genomic products based on defined features. This analysis consists of three facets: molecular functions (MF), biological processes (BP) and cellular components (CC). The Kyoto Encyclopedia of Genes and Genomes (KEGG) is a database resource for understanding high-level biological functions and utilities. These analyses and annotations are based on the DAVID database (https://david.ncifcrf.gov/), which provides a comprehensive set of functional annotation tools for investigators to

explore and understand the biological meaning underlying particular gene lists. In this study, both GO and KEGG analyses of DEGs were performed with a criterion false discovery rate (FDR) < 0.05.

## Protein–protein interaction (PPI) network construction and hub gene analysis

In order to analyze the connections among the proteins encoded by identified DEGs, DEGs were uploaded to Search Tool for the Retrieval of Interacting Genes (STRING, https://string-db.org/), a database of known and predicted protein-protein interactions, and the results with a minimum interaction score of 0.4 were visualized in Cytoscape. Furthermore, CytoHubba, a Cytoscape plugin app, providing a user-friendly interface to explore important nodes in biological networks, was utilized with the maximal clique centrality (MCC) method to explore the PPI network for hub genes.

## DEGs validation

Another dataset, GSE14323, was used to confirm the validity and disease relevance of identified DEGs. A heat map of the expression of 25 hub genes was developed using the HemI1.0.3.3 software. Statistical difference analysis between the liver cirrhosis group (LC) and normal control group (NC) was performed via student's $t$-test using SPSS V20.0. $P <$ 0.05 was considered statistically significant. As activation of hepatic stellate cells (HSCs) is considered as a central driver of liver fibrosis, we used a human HSC cell line—LX2 treated with TGF-β1 to represent this activation stage. An expression of 25 hub genes was performed by quantitative real-time PCR assay compared with normal control.

## Cell culture and treatment

The LX2 cell line was purchased from Procell Life Science & Technology (Wuhan, China), cultured with Dulbecco Modified Eagle Medium (DMEM)-high glucose supplemented with 10% fetal bovine serum (FBS) and antibiotics (100 U/mL penicillin-G and 100 μg/mL streptomycin), and incubated at 37 °C in 5% CO2 and 95% humidified air. The LX2 cells were seeded in a 10-cm culture dish at a density of $1 * 10^6$ for 6 h. After attachment, the LX2 cells were treated with TGF-β1 (R&D systems, Catalog #240-B/CF) at 10 ng/ml concentration or left untreated as normal control for 24 h. Then RNA and proteins were isolated for further use.

## Western blot assay

LX2 cells total protein were extracted with ice-cold RIPA lysis buffer. Protein concentration was determined using the BCA Protein Assay Kit (Thermo Fisher Scientific, Waltham, MA, USA). Quantified proteins were separated on SDS-PAGE and transferred onto PVDF membranes (Millipore Corporation, Burlington, MA, USA). After blocking, membranes were incubated with anti-αSMA (1:20,000, ab124964, Abcam, UK) at 4 °C overnight. Then, membranes were washed with TBST and incubated with secondary antibodies for 2 h at room temperature. The anti-GAPDH (1:1,000, CST) was set as internal control. Protein bands were visualized by using ECL equipment (Pierce Chemical, Waltham, MA, USA).

## Quantitative real-time PCR assay

RNA was extracted from cell line LX2 by Trizol reagent (Takara, Kusatsu, Japan) by following the manufacturer's instructions. The cDNAs were synthesized with a commercial kit (Takara, Japan). Gene expressions were measured by real-time PCR with CFX Connect™ Real-Time PCR System (Bio-Rad, Hercules, CA, USA). GAPDH was used as an internal control and the relative expression levels of mRNA were calculated using the $2^{-\Delta\Delta Ct}$ method. The primer pairs used in the experiments are listed in Data S1.

## Prediction of therapeutic agents and target genes

To discover potential anti-fibrotic agents, the identified 185 DEGs were queried using the Connectivity Map online tool (L1000 platform; https://clue.io/l1000-query). This tool compares queried signatures with a gene expression profile database of several cell lines after treatment with more than 2,000 compounds, most of which are FDA approved. Drugs whose signatures were in opposition to the disease signature were assumed to have therapeutic potential.

# RESULTS

## Identification of 185 conserved DEGs

As shown in Fig. 1, each dataset was initially analyzed separately to identify DEGs unique to fibrosis of a given origin. 1,563 DEGs were identified in GSE6764 (HCV), 243 DEGs in GSE49541 (NAFLD), and 1,396 DEGs in GSE84044 (HBV) (Data S2). 185 DEGs overlapped across all three datasets, suggesting that these fibrosis-related DEGs may be conserved regardless of disease etiology. Among these 185 DEGs, 174 were up-regulated while only 11 were down-regulated. Interestingly, although the number of DEGs in NAFLD related dataset is relatively small, 94.7% of the 243 DEGs intersect with other datasets.

## Functional enrichment analysis of DEGs

In order to compare the differences in gene function among these 3 datasets, GO and KEGG analyses were performed on each dataset (Data S3) and top 10 significant GO_BP and KEGG pathways are shown in Tables 2 and 3, respectively. Unexpectedly, the cell cycle pathway was uniquely enriched in the HBV-related dataset, ranking third among all KEGG pathways for this dataset. When compared with a non-viral fibrosis dataset (GSE49541), those datasets in which fibrosis was of viral origin (GSE6764 and GSE84044) contained DEGs enriched for immune-inflammatory responses, consistent with the distinct role of immunological responses in the initiation and control of local disease in affected individuals.

Next, GO and KEGG analyses were performed on the 185 common DEGs. The GO analysis revealed that most of the proteins encoded by these DEGs were extracellular matrix proteins located in the extracellular space (Fig. 2). The molecular functions (MF) enriched in this dataset were primarily associated with platelet-derived growth-factor binding and extracellular matrix structural constitution, while the enriched biological processes (BP) were primarily those associated with extracellular matrix organization and cell adhesion (Fig. 2). The KEGG analysis revealed that the primary enriched signaling
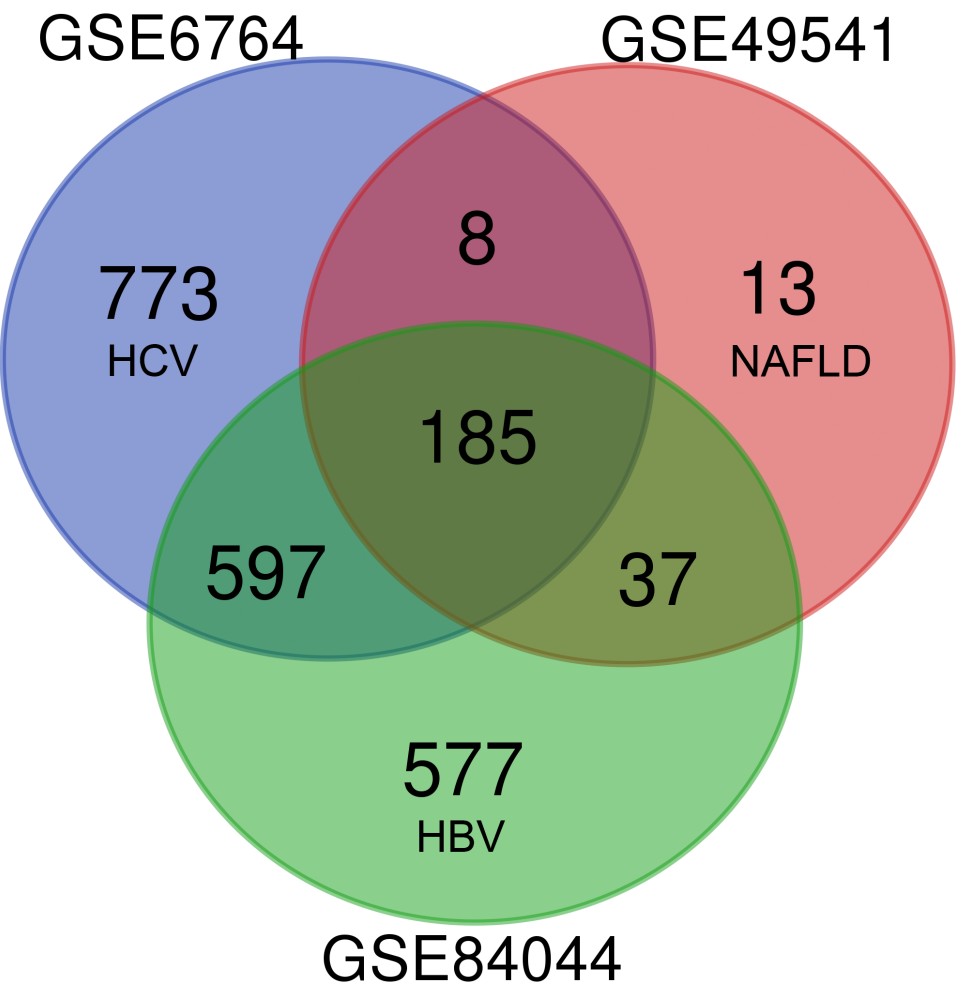

**Figure 1** **Venn diagram of DEGs from the three cohort profile sets (GSE6764, GSE49541, GSE84044), generated using an online tool.** Each colored circle represents a different dataset, and areas of overlap indicate shared DEGs. Statistically significant DEGs were defined based on *adj.p* < 0.05 and [FC] > 1.5 as the cut-off criteria.

pathways were those associated with ECM-receptor interaction, protein digestion and absorption, focal adhesion, and the PI3K-Akt signaling pathway (Fig. 3). Together, these shared DEGs highlight the central roles of cell–cell adhesion and ECM dysregulation in the development of fibrosis, regardless of the etiological origin of the disease.

## PPI network construction and hub gene identification

To better understand which of these shared DEGs were most likely to be the key genes most essential for the development of fibrosis, a PPI network for these 185 common DEGs was built with 105 nodes and 275 edges. 80 of the 185 DEGs were not included in the PPI network (Fig. 4), as interaction score of these 80 genes were less than 0.4. Among the 105 genes in the PPI network, the top 25 genes according to the MCC method were selected using the CytoHubba plugin and are sequentially ordered as follows: COL1A2,

**Table 2  Top 10 GO_BP terms in each datasets ordered by FDR.** Count: number of genes enriched in the corresponding pathway; FDR, false discovery rate.

| GO ID | Biological process | Count | FDR |
|-------|-------------------|-------|-----|
| GSE6764-HCV | | | |
| GO:0030198 | Extracellular matrix organization | 59/196 | 8.35E−17 |
| GO:0060337 | Type I interferon signaling pathway | 33/64 | 3.82E−16 |
| GO:0006955 | Immune response | 88/421 | 7.84E−15 |
| GO:0051607 | Defense response to virus | 47/165 | 9.18E−12 |
| GO:0060333 | Interferon-gamma-mediated signaling pathway | 30/71 | 1.73E−11 |
| GO:0007155 | Cell adhesion | 79/459 | 2.87E−08 |
| GO:0007165 | Signal transduction | 144/1,161 | 9.27E−06 |
| GO:0050900 | Leukocyte migration | 31/122 | 1.34E−05 |
| GO:0045071 | Negative regulation of viral genome replication | 17/40 | 2.56E−05 |
| GO:0042493 | Response to drug | 53/304 | 5.86E−05 |
| GSE49541-NAFLD | | | |
| GO:0030198 | Extracellular matrix organization | 29/196 | 6.55E−18 |
| GO:0007155 | Cell adhesion | 37/459 | 7.26E−15 |
| GO:0030574 | Collagen catabolic process | 13/64 | 6.27E−08 |
| GO:0030199 | Collagen fibril organization | 9/39 | 5.87E−05 |
| GSE84044-HBV | | | |
| GO:0006955 | Immune response | 87/421 | 2.1693E−16 |
| GO:0007155 | Cell adhesion | 81/459 | 7.1809E−11 |
| GO:0030198 | Extracellular matrix organization | 46/196 | 2.2322E−09 |
| GO:0070374 | Positive regulation of ERK1 and ERK2 cascade | 38/175 | 2.6745E−06 |
| GO:0070098 | Chemokine-mediated signaling pathway | 23/71 | 3.4269E−06 |
| GO:0002250 | Adaptive immune response | 33/148 | 1.927E−05 |
| GO:0060326 | Cell chemotaxis | 21/65 | 2.1359E−05 |
| GO:0006954 | Inflammatory response | 58/379 | 9.3052E−05 |
| GO:0002548 | Monocyte chemotaxis | 16/42 | 0.00015641 |
| GO:0030574 | Collagen catabolic process | 19/64 | 0.00052759 |

COL1A1, COL6A3, COL3A1, COL5A2, COL5A1, COL4A1, COL4A2, COL4A3, COL4A4, DCN, COL14A1, LUM, COL15A1, THBS2, FBN1, ITGB8, CDH11, ADAMTS2, CTGF, VCAN, PCOLCE2, SPP1, VWF, CTSK (Fig. 5). These 25 genes were deemed to be the hub genes and were those genes most likely to be essential for fibrogenesis. Most genes encode ECM components, including COL1A2, DCN, and FBN1. Other hub genes play known roles in ECM structural regulation (THBS2, ITGB8, VWF), while some are associated with ECM degradation (ADAMTS2, PCOLCE2, CTSK). This finding is consistent with known fibrogenic mechanisms, and suggests key potential drug targets that are most likely to have effective anti-fibrotic activity when disrupted.

## Hub gene validation

In order to extend and validate our findings in a distinct model of human liver fibrosis, these top 25 hub genes were validated in the GSE14323 dataset, in which liver cirrhotic (LC) and normal control tissues (NC) were selected for analysis. Figure 6 displays a

**Table 3  Top 10 KEGG pathways in each dataset ordered by FDR.**

| KEGG ID | Pathway | Count | FDR |
|---------|---------|-------|-----|
| GSE6764-HCV | | | |
| hsa04514 | Cell adhesion molecules (CAMs) | 44/142 | 2.43E−10 |
| hsa05332 | Graft-versus-host disease | 18/33 | 4.53E−07 |
| hsa05330 | Allograft rejection | 18/37 | 4.39E−06 |
| hsa04940 | Type I diabetes mellitus | 19/42 | 6.01E−06 |
| hsa05416 | Viral myocarditis | 22/57 | 8.34E−06 |
| hsa04510 | Focal adhesion | 46/206 | 1.08E−05 |
| hsa04512 | ECM-receptor interaction | 27/87 | 2.20E−05 |
| hsa04612 | Antigen processing and presentation | 25/76 | 2.37E−05 |
| hsa05164 | Influenza A | 39/174 | 1.64E−04 |
| hsa05168 | Herpes simplex infection | 40/183 | 2.22E−04 |
| GSE49541-NAFLD | | | |
| hsa04512 | ECM-receptor interaction | 17/87 | 3.17E−10 |
| hsa04510 | Focal adhesion | 21/206 | 6.22E−08 |
| hsa04974 | Protein digestion and absorption | 13/88 | 1.32E−05 |
| hsa04151 | PI3K-Akt signaling pathway | 22/345 | 9.83E−05 |
| hsa05146 | Amoebiasis | 11/106 | 0.007509 |
| GSE84044-HBV | | | |
| hsa04512 | ECM-receptor interaction | 28/87 | 7.34E−07 |
| hsa05323 | Rheumatoid arthritis | 27/88 | 5.01E−06 |
| hsa04110 | Cell cycle[a] | 32/124 | 1.4E−05 |
| hsa04672 | Intestinal immune network for IgA production | 18/47 | 9.61E−05 |
| hsa05150 | Staphylococcus aureus infection | 19/54 | 0.000169 |
| hsa05222 | Small cell lung cancer | 23/85 | 0.001041 |
| hsa04151 | PI3K-Akt signaling pathway | 56/345 | 0.001159 |
| hsa04510 | Focal adhesion | 39/206 | 0.00179 |
| hsa05166 | HTLV-I infection | 45/256 | 0.00197 |
| hsa04974 | Protein digestion and absorption | 23/88 | 0.001973 |

**Notes.**
[a]Pathway is unique in the corresponding dataset.

heatmap of GSE14323 expression profile data. This expression profile was consistent with the overlapping DEGs identified in the initial three datasets, with 23 of these 25 hub DEGs being up-regulated in cirrhotic patients. Statistical analysis of these genes in the validation dataset is shown in Fig. 7. Differences for all the hub genes between the LC and NC groups were statistically significant with the exception of ITGB8.

A cell model of liver fibrosis was also constructed to validate these 25 hub genes. When treated with TGF-β1, the LX2 cells extended more tentacles and expressed more a-SMA protein (One of the markers of hepatic stellate cell activation) (Fig. 8), indicating the cell model was successfully established. Figure 9 displays statistical analysis of 25 hub genes relative expression to GAPDH, 13 of these 25 hub genes was up-regulated significantly, which is consistent with the trend of GEO datasets in this study. However, 4 genes (LUM, THBS2, ITGB8 and SPP1) was down-regulated in TGF-β1 treated cells. The expression
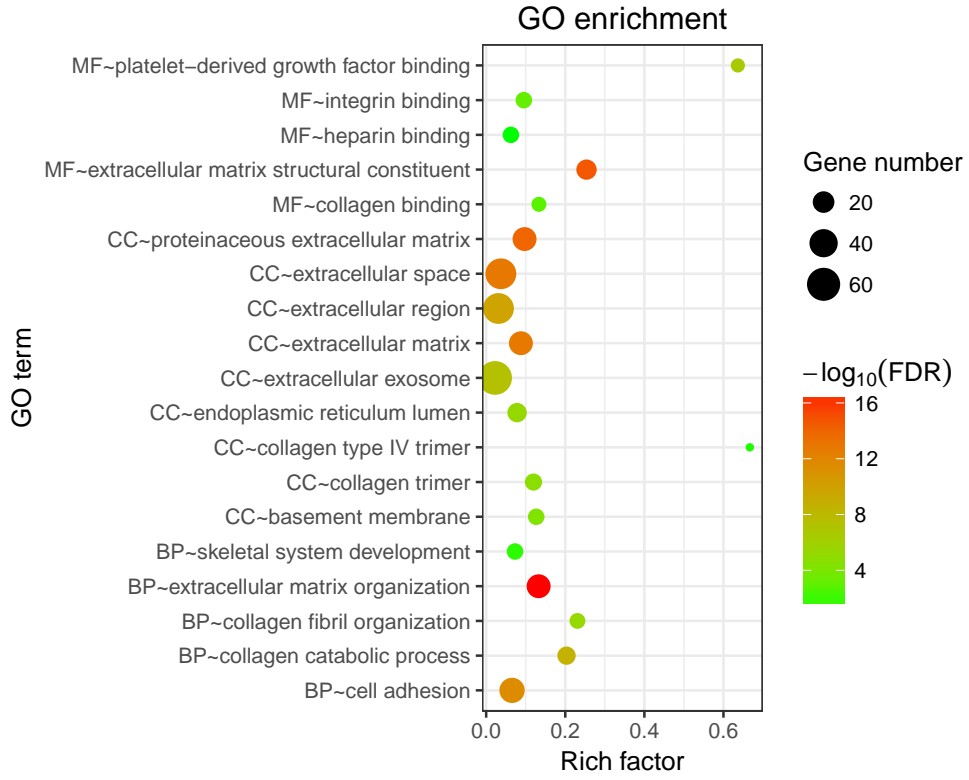

**Figure 2    GO enrichment analyses of 158 common DEGs.** The top 10 terms in each GO category (MF, molecular function; CC, cellular components; BP, biological processes).

trend of some genes was inconsistent with expectations, likely because TGF-β-induced hepatic stellate cell activation does not represent the entire activated form or that part of the gene is expressed by other cells such as hepatic parenchymal cells and Kupffer cells.

## Prediction of potential therapeutic agents and targets

Given the role of 185 DEGs in fibrogenesis, we next wanted to probe for potential therapeutic compounds that might best be suited to target these genes in order to achieve a beneficial therapeutic outcome. To that end the connectivity map L1000 platform, which compiles gene expression profiles associated with a wide range of therapeutic compounds, was used to search for drugs with the potential for therapeutic repurposing as a means of treating liver fibrosis (Data S4). The top 10 compounds are shown in Table 4, and when sequentially ordered by median_score are: Prometon, MK-212, Evodiamine, Zosuquidar, CAY-10415, Caffeic-acid, Budesonide, Rilmenidine, Afatinib, Desloratadine. Target genes corresponding to each compound (with the exception of prometon) were also listed. Among these target genes, three (HTR2B, ABCB1 and ALOX5) were significantly up-regulated in HBV and HCV datasets, and the mRNA expression levels in each dataset were listed in Table 5. Together, these compounds and target genes provide a promising list for researchers or companies interested in conducting pre-clinical research into the mechanisms of and treatments for fibrosis both *in vitro* and *in vivo*.

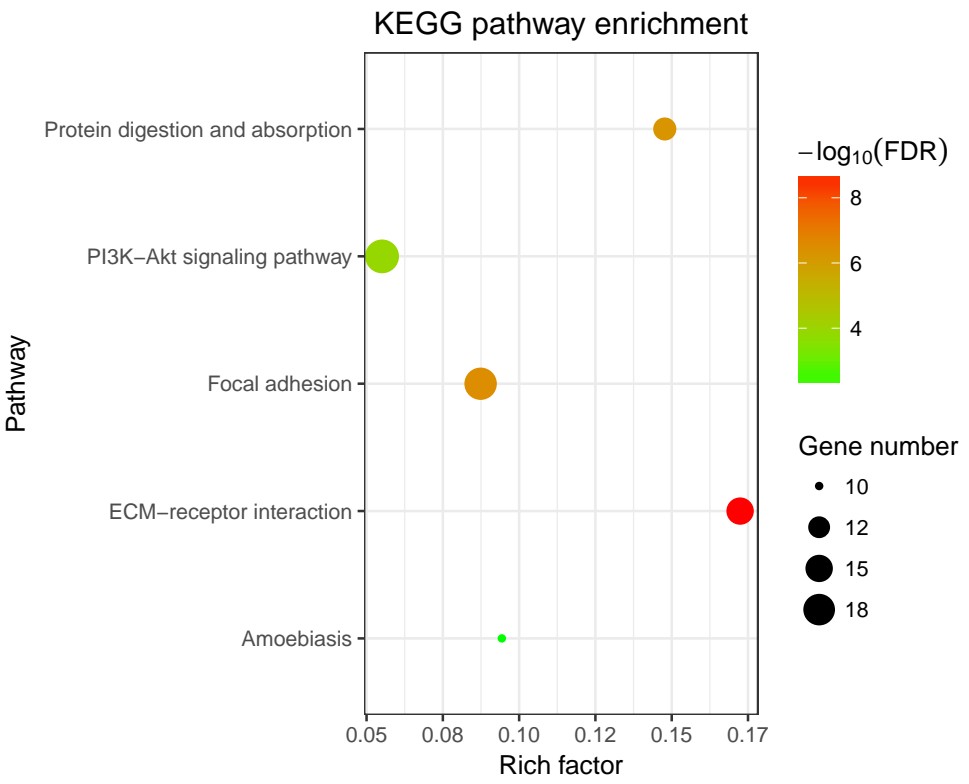

**Figure 3 KEGG enrichment analyses of 158 common DEGs.** All significant KEGG pathways. GO and KEGG analysis was performed using the DAVID online tool with the cutoff criteria of FDR < 0.05. The color of each bubble represents the FDR for that term, with red representing greater significance. The rich factor refers to the proportion of enriched genes for each term.

## DISCUSSION

Globally, HBV, HCV, and NAFLD are the most three common causes of liver fibrosis (*Altamirano-Barrera, Barranco-Fragoso & Méndez-Sánchez, 2017*). In the present study, we integrated datasets that were focused on these three most common causes of liver fibrosis, and in so doing we were able to identify different and common signaling pathways for the fibrogenesis.

Transition of hepatic stellate cells (HSCs) from a quiescent to an activated state is a sign of the onset of liver fibrosis, and this process is controlled by E-type cyclins (CcnE1, CcnE2) and their associated cyclin-dependent kinase 2 (Cdk2) (*Nevzorova et al., 2012*; *Ohtsubo et al., 1995*). Cyclin E-Cdk2 has long been considered an essential master regulator of progression through the G1 phase of the cell cycle (*Hwang & Clurman, 2005*). According to our KEGG pathway enrichment results, the cell cycle pathway is uniquely enriched in HBV-related fibrosis dataset, with CcnE2 being significantly up-regulated only in this HBV dataset, but not in the HCV or NAFLD datasets (Table 5). Therapeutic targeting of Cyclin E1 via RNAi has been shown to have robust anti-fibrotic activity in mice (*Bangen et al., 2017*), and if this technology can be applied clinically in future, we predict that it will be most effective in those patients with chronic hepatitis B.

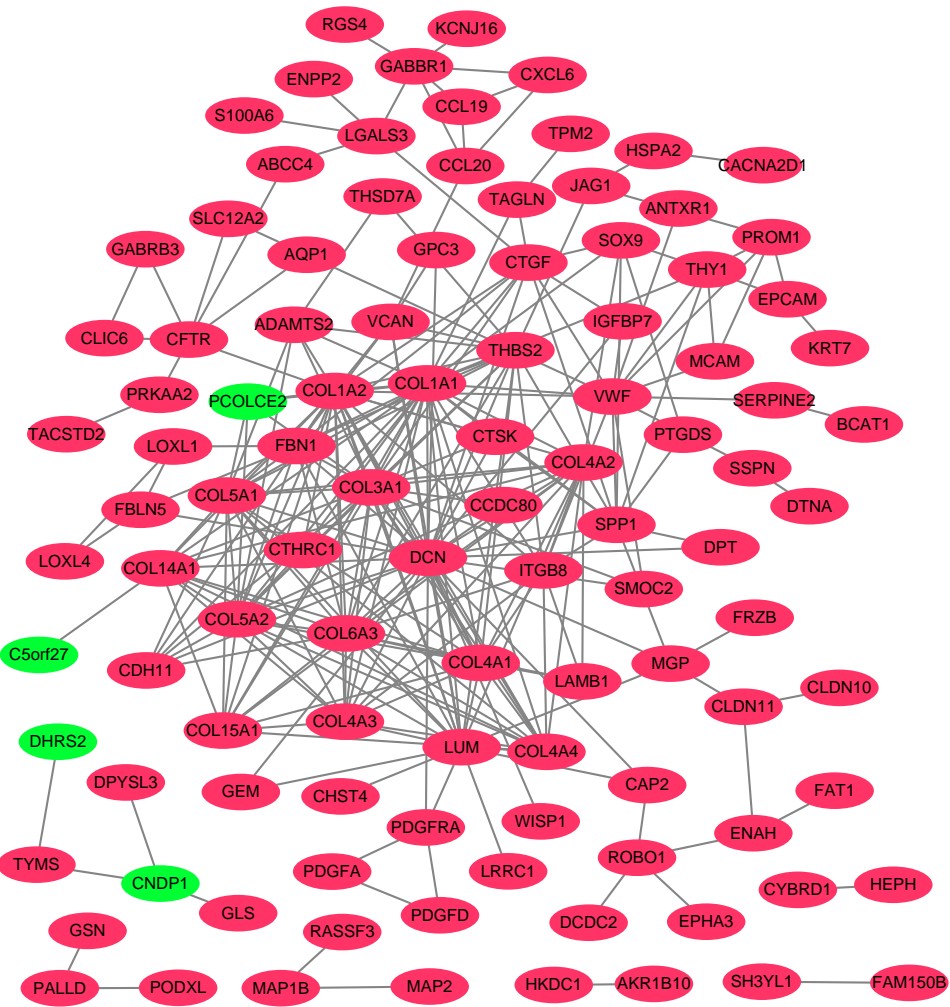

**Figure 4  Protein–protein interaction (PPI) network complex.** Using the STRING online database, a total of 105 DEGs (101 up-regulated genes shown in red and four down-regulated genes shown in green) were filtered into a DEG PPI network complex. The gray line between two proteins means an interaction score no less than 0.4, and the more interactions with other proteins, the more important this protein is.

Both HBV and HCV are non-cytopathic viruses, and liver damage in infected individuals is mainly caused by an inflammatory immune response aimed at eliminating the virus (*Guidotti & Chisari, 2006*), with such persistent inflammation leading to liver fibrosis (*Protzer, Maini & Knolle, 2012*). In contrast to such fibrosis of viral origin, the production of reactive oxygen species (ROS) and resulting oxidative stress is thought to be a critical factor in NAFLD-associated fibrosis. Although NAFLD is always accompanied by an inflammatory reaction with variations in levels of pro-inflammatory cytokines (*Cai et al., 2005*), the immune-inflammatory response is not significantly enriched in this NAFLD-related dataset, whereas it is clearly enriched in the virus-related datasets.

By using a PPI network analysis, we identified 25 hub genes, some of which have been previously reported including LUM, THBS2, FBN1 (*Lou et al., 2017*), COL6A3,

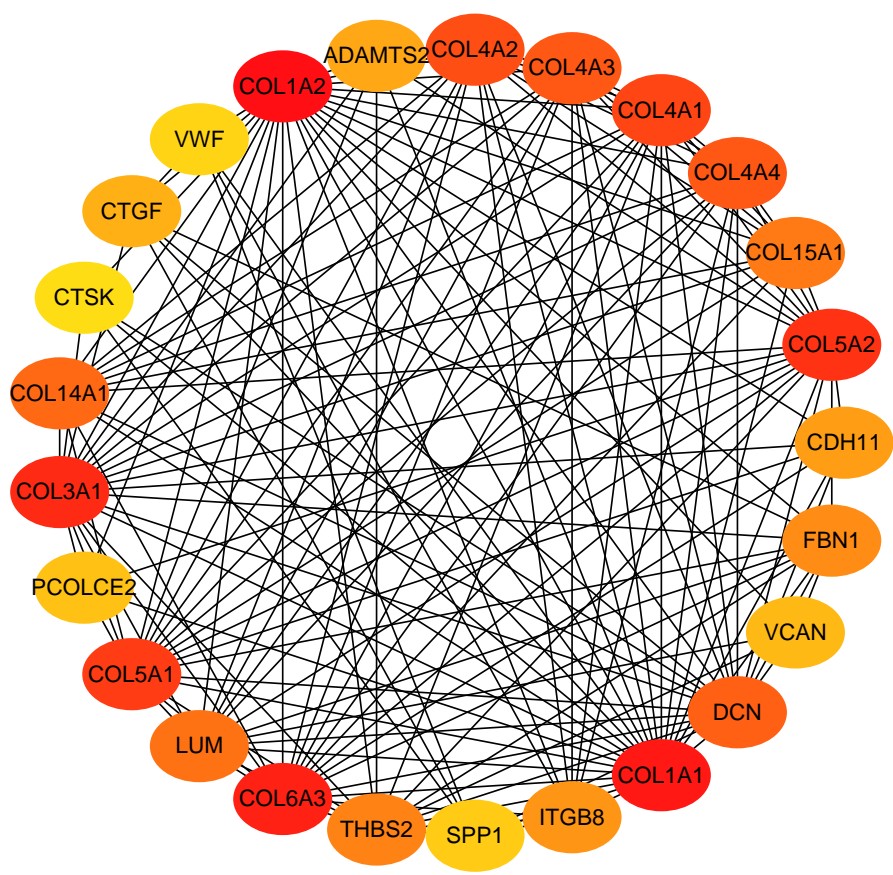

**Figure 5  Top 25 hub gene network.** The top 25 genes derived from the MMC method were chosen using the CytoHubba plugin. Advanced ranking is represented by a redder color.

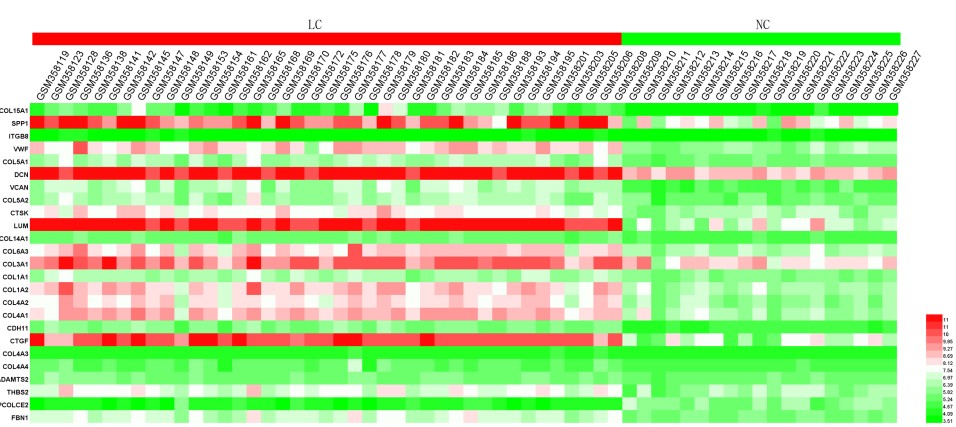

**Figure 6  Heatmap of the expression of the 25 hub genes in the GSE14323 validation dataset.** Red color, up-regulated; green color, down-regulated.

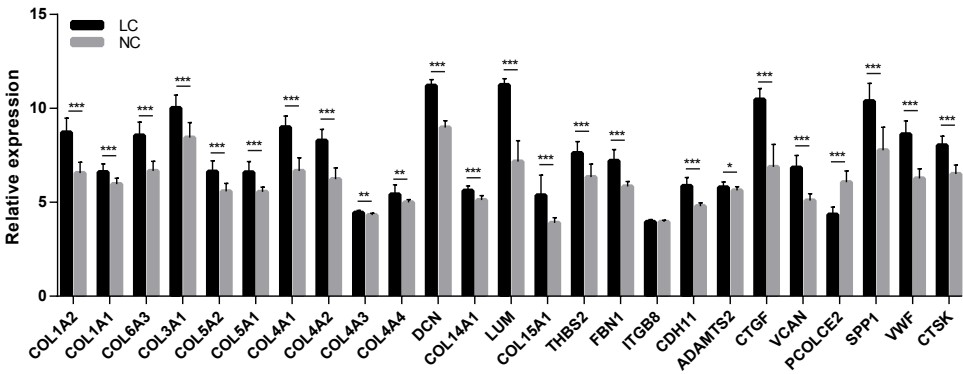

**Figure 7** **Statistical analysis of the expression of 25 hub genes in GSE14323.** The differences in expression of all hub genes between the liver cirrhosis (LC) group and the normal control (NC) group were statistically significant with the exception of ITGB8. LC, liver cirrhosis; NC, normal control. $*p < 0.05$, $**p < 0.01$, $***p < 0.001$.

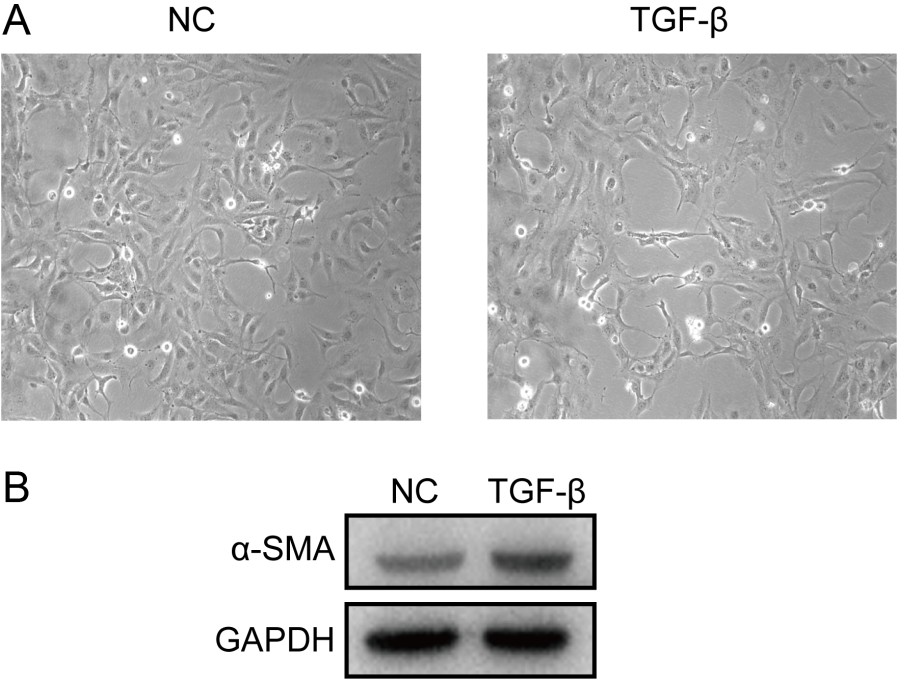

**Figure 8** **Successful construction of liver fibrosis cell model.** After treated with TGF-β1 for 24 h the LX2 cells morphology became irregular and extended more tentacles (A), and expressed more a-SMA protein (one of the markers of hepatic stellate cell activation) determined by western blot (B).

and COL1A1 (*Chen et al., 2017*). These 25 hub genes are crucial to fibrogenesis, and are expressed regardless of etiology. The development of anti-fibrotic drugs should therefore focus on these genes as targets. Connective tissue growth factor (CTGF), one of the 25 identified hub genes, is expressed at very low levels in normal liver tissue but is significantly upregulated in fibrotic liver tissue. Recently, a clinical trial of patients with

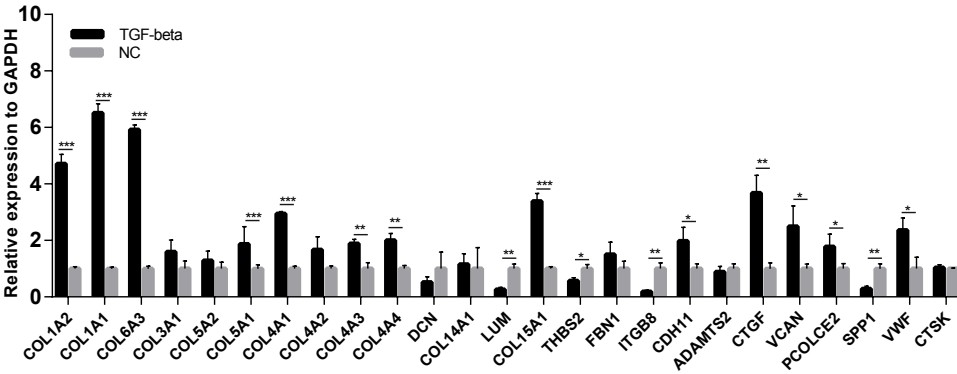

**Figure 9** **mRNA expressions of 25 hub genes in cell model by qPCR.** When LX2 cells were activated by TGF-β1 13 genes significantly up-regulated which is consistent with the trend of microarray data in this study. four genes (LUM, THBS2, ITGB8 and SPP1) was down-regulated.

**Table 4** **Top 10 compounds predicted to have activity against liver fibrosis as predicted via connectivity map.**

| ID | Median_Score | Name | Description | Target |
|---|---|---|---|---|
| BRD-K99029477 | −93.11 | Prometon | Photosynthesis inhibitor | |
| BRD-K19554809 | −88.43 | MK-212 | Serotonin receptor agonist | HTR2A, HTR2B[a], HTR2C |
| BRD-A68631409 | −85.73 | Evodiamine | ATPase inhibitor, TRPV agonist | TRPV1 |
| BRD-K70557564 | −85.48 | Zosuquidar | P glycoprotein inhibitor, P glycoprotein modulator | ABCB1[a], ABCB4 |
| BRD-A61858259 | −83.89 | CAY-10415 | Insulin sensitizer | INS |
| BRD-K09900591 | −81.7 | Caffeic-acid | Lipoxygenase inhibitor, HIV integrase inhibitor, NFkB pathway inhibitor, nitric oxide production inhibitor, PPAR receptor modulator, tumor necrosis factor production inhibitor | ALOX5[a], MIF, RELA, TNF |
| BRD-A82238138 | −81.69 | Budesonide | Glucocorticoid receptor agonist, glucocordicoid receptor antagonist, immunosuppressant | NR3C1 |
| BRD-K52080565 | −80.2 | Rilmenidine | Adrenergic receptor agonist, imidazoline receptor agonist | NISCH |
| BRD-K66175015 | −75.46 | Afatinib | EGFR inhibitor, receptor tyrosine protein kinase inhibitor, tyrosine kinase inhibitor | EGFR, ERBB2, ERBB4 |
| BRD-K82357231 | −74.73 | Desloratadine | Histamine receptor antagonist | HRH1 |

**Notes.**
[a]Targets were matched to DEGs in selected datasets.

**Table 5** **mRNA expression levels of selected genes in each datasets.** logFC, log2 (Fold Change); Genes with an expression level of |logFC| > 0.585 and $adj.P.Val < 0.05$ were deemed to be the DEGs in this study.

| Gene symbol | GSE6764 (HCV) | | GSE49541 (NAFLD) | | GSE84044 (HBV) | |
|---|---|---|---|---|---|---|
| | logFC | $adj.P.Val$ | logFC | $adj.P.Val$ | logFC | $adj.P.Val$ |
| HTR2B | 1.24 | 1.63E−02 | 0.33 | 2.26E−01 | 0.74 | 2.61E−06 |
| ABCB1 | 0.75 | 2.24E−02 | 0.26 | 1.75E−01 | 0.76 | 9.61E−08 |
| ALOX5 | 0.99 | 2.88E−02 | 0.23 | 2.72E−01 | 1.06 | 7.79E−09 |
| CcnE2 | −0.39 | 1.09E−01 | 0.46 | 4.36E−02 | 0.96 | 9.91E−07 |

HBV-associated liver fibrosis was completed utilizing a human monoclonal antibody reactive to CTGF (NCT01217632). Although this clinical trial was terminated due to an unexpected prominent single-agent effect of entecavir in this patient population, it did show promise.

In this study, we were able to screen for and identify 10 compounds that may have therapeutic activity against liver fibrosis. Among these compounds, evodiamine, caffic-acid, and budesonide have all been shown to be effective in animal models or clinical trials (*Alferink et al., 2017*; *Silveira & Lindor, 2014*; *Yang et al., 2018*; *Yang et al., 2017*). Four agents, MK-212, CAY-10415, afatinib, and desloratadine, have not been tested *in vivo*, but compounds targeting the same molecules as these four agents have been reported to have the potential to ameliorate liver fibrosis (*Boettcher et al., 2012*; *Ebrahimkhani et al., 2011*; *Fuchs et al., 2014*; *Kennedy et al., 2018*). Prometon, Zosuquidar, and Rilmeidine have not yet been reported to have relationship with fibrosis.

Among the target genes corresponding to these compounds, we found that HTR2B, ABCB1, and ALOX5 were significantly up-regulated in HBV and HCV related liver fibrosis datasets (Data not show). Stimulation of the 5-hydroxytryptamine 2B receptor (HTR2B) on HSCs by serotonin is required to negatively regulate hepatocyte regeneration, and antagonism of HTR2B has been shown to attenuate fibrogenesis and improve liver function in disease models in which fibrosis was pre-established and progressive (*Ebrahimkhani et al., 2011*). Interestingly, MK-212, an HTR2B agonist, showed a negative liver fibrosis gene expression profile suggesting potential as an anti-fibrotic agent, although formal experimental testing is needed. Arachidonate 5-lipoxygenase (ALOX5) plays a role in the synthesis of leukotrienes from arachidonic acid, and inhibition of the ALOX5 pathway markedly reduces the number of Kupffer cells in culture and attenuates inflammation and fibrosis in experimental liver disease (*Titos et al., 2003*). Recently, a clinical study revealed that frequent coffee consumption was inversely correlated with liver stiffness (*Alferink et al., 2017*), with suggestions that the underlying mechanism may be one related to the inhibition of TGF-β1/Smad3 signaling and the induction of autophagy in HSCs in response to caffeic acid (*Yang et al., 2017*). As an inhibitor of ALOX5, caffeic acid may thus be able to attenuate liver fibrosis via this ALOX5 (*Sud'ina et al., 1993*) pathway.

ATP Binding Cassette Subfamily B Member 1 (ABCB1) is known for encoding P glycoprotein, which is responsible for decreased drug accumulation in multidrug-resistant cells and often mediates the development of resistance to anticancer drugs, such as Zosuquidar mentioned above. However, there are currently no studies reporting that ABCB1, P glycoprotein or Zosuquidar is associated with liver fibrosis. Some studies have reported P glycoprotein was increased in rat activated HSC (*Hannivoort et al., 2008*), and its activity was increased by TGF-β (*Baello et al., 2014*) and endoplasmic reticulum stress (*Ledoux et al., 2003*), which are considered to be effective activators of HSC. Combining the findings of our research, we infer that ABCB1 might be a novel therapeutic target to liver fibrosis, although this hypothesis need to be verified in further study.

## CONCLUSIONS

Our study integrated three liver fibrosis datasets, each with fibrosis of a different etiology (HBV, HCV and NAFLD). Through the functional analysis of identified DEGs, we revealed that cell cycle is a pathway uniquely enriched in HBV related dataset and immune-inflammatory response is clearly enriched in the virus-related datasets. We further identified 25 key hub genes, the majority of which were linked to ECM regulation, highlighting the central processes common to all causes of fibrogenesis, offering valuable insights into the conserved nature of fibrotic signaling. Based on the 185 DEGs, we were additionally predicted 10 compounds, especially Zosuquidar and corresponding gene target ABCB1, may have anti-fibrotic activity. While further experiments will be needed to validate these findings, this successful compound screening effort suggests that it may be possible to repurpose extant drugs to more readily treat liver fibrosis.

### Funding

This work was supported by the National Science and Technology Major Project of China (2018ZX10302206-003, 2017ZX10202203-007, 2017ZX10202203-008). The funders had no role in study design, data collection and analysis, decision to publish, or preparation of the manuscript.

### Grant Disclosures

The following grant information was disclosed by the authors:
National Science and Technology Major Project of China: 2018ZX10302206-003, 2017ZX10202203-007, 2017ZX10202203-008.

### Competing Interests

The authors declare there are no competing interests.

### Author Contributions

- Zhu Zhan conceived and designed the experiments, performed the experiments, analyzed the data, contributed reagents/materials/analysis tools, prepared figures and/or tables, authored or reviewed drafts of the paper, approved the final draft.
- Yuhe Chen and Yuanqin Duan performed the experiments, prepared figures and/or tables, approved the final draft.
- Lin Li analyzed the data, prepared figures and/or tables, approved the final draft.
- Kenley Mew authored or reviewed drafts of the paper, approved the final draft.
- Peng Hu and Hong Ren conceived and designed the experiments, contributed reagents/materials/analysis tools, approved the final draft.
- Mingli Peng conceived and designed the experiments, contributed reagents/materials/-analysis tools, authored or reviewed drafts of the paper, approved the final draft.

## Microarray Data Deposition

The following information was supplied regarding the deposition of microarray data:

Data is available at the Gene Expression Omnibus database, accession numbers: GSE6764, GSE49541, GSE84044, GSE14323.

## Data Availability

The raw measurements are available in Data S1–S4.

## Supplemental Information

Supplemental information for this article can be found online at http://dx.doi.org/10.7717/peerj.6645#supplemental-information.

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
