# Peer review of "Identification of key genes, pathways and potential therapeutic agents for liver fibrosis using an integrated bioinformatics analysis"

_PeerJ, doi:10.7717/peerj.6645_

## Round 0.1 · original submission · Major Revisions

Please address all the critiques raised by both reviewers and revise the manuscript accordingly.

Reviewer 1 ·

Basic reporting

The manuscript is clear and unambiguous. Professional English is used throughout.

Sufficient literature is cited.

The article structure, figures, and tables have a professionally drafted.

The manuscript is self-contained with relevant results to a hypothesis.

Experimental design

The experimental design is within the aims and scope of the journal.

The research question is well defined and straightforward.

A thorough investigation is performed.

The methods are satisfactorily described with sufficient detail.

Validity of the findings

The study impactful with novel implications.

The data provided is robust and statistically sound.

The conclusion is well stated and well linked with the original research question.

Additional comments

The authors have used bioinformatics analysis to identify gene, pathways and drugs involved in liver fibrosis. However, the manuscript can only be accepted for publication after clarification of following points.
1. CcnE2 is significantly upregulated in the HBV dataset. Please mention at what state of cell cycle CcnE2 is present and expressed? What are the potential inhibitors for CcnE2? Also what is the primary function of CcnE2?
2. What is the molecular structure of Zosuquidar?
3. How the Zosuquidar interacts with ABCB1? Please elaborate briefly?
4. Only three datasets are been chosen for the study based on the rationale of fibrosis of a different etiology. Are there more datasets like these are available? If yes, what is the main reason of choosing only these datasets? Please elaborate.
5. Figure 4, please clarify the gene marked genes in the figure legends.

Reviewer 2 ·

Basic reporting

In this work, the authors used bioinformatics tools to analyze three gene expression profiles of liver fibrosis from GEO database and identified key genes in the process along with therapeutic agents that may have antifibrotic activity.

The manuscript lacks on sufficient background information and literature references as noted in general comments.

Experimental design

The aim of the study is well-defined and the methods used are appropriate.

Validity of the findings

Conclusions are well stated and supported by the results shown.

Additional comments

1) Introduction needs improvement and should be rewritten as sufficient background covering previous bioinformatics work is not provided to understand the significance of the work. For example, lines 78-81, “However, these approaches have been limited in their ability to identify the key genes regulating the disease as a whole. Moreover, the results of these studies are often inconsistent due to different etiologies of liver fibrosis”. Authors should provide information on key genes identified in previous studies, and in the discussion section, elaborate on the similarities/differences in the hub genes identified in this manuscript to those already reported in literature.
2) On lines 222-224, authors suggest that the 25 hub genes could be potential drug targets for antifibrotic activity. However, none of the hub genes seems to be targeted by the top 10 identified compounds (reported in Table 4) that authors suggest could be potential antifibrotic therapeutic agents. How do authors reconcile this difference?
3) On lines 274-275, “…with CcnE2 being significantly up-regulated only in this HBV dataset, but not in the HCV or NAFLD datasets”. This statement is not supported by the data shown in the manuscript.
4) On lines 285-286, “…the degree of inflammation is typically less severe than that caused by HBV/HCV according to our results”. It is not clear how the results in the paper relate to degree of inflammation. Can the authors elaborate on this?
5) On lines 292-293, “but the other agents targeting similar molecules have the potential to ameliorate liver fibrosis”. It is not clear what “other agents” or “similar molecules” the authors are referring to here.
6) Supplementary data 2-4 have been shared for review but have not been referenced in the paper anywhere.

---

## Round 0.2 · accepted · Accept

Both reviewers were satisfied by your responses to their critiques and by the modifications in the revised version of the manuscript. Therefore, the manuscript is acceptable now.

# Reviewer 1 ·

Basic reporting

No Comments.

Experimental design

No comments.

Validity of the findings

No comments.

Additional comments

The authors have done the recommended changes and modifications in the newer version of the manuscript. Considering these change I now can recommend the manuscript for publication.

Reviewer 2 ·

Basic reporting

no comment

Experimental design

no comment

Validity of the findings

no comment

Additional comments

Authors have addressed all my concerns and the manuscript is now suitable for publication.